# TSLPR deficiency attenuates AHR independently of eosinophilia and mucus secretion in a chronic HDM mouse model of allergic asthma

Latifa Koussih[1,2], Sina Taefehshokr[1], Lianyu Shan[1], Sujata Basu[3], Andrew Halayko[3], Bouchaib Lamkhioued[4], Abdelilah S. Gounni[1]*

1 Department of Immunology, Rady Faculty of Health Sciences, Max Rady College of Medicine, University of Manitoba, Winnipeg, Manitoba, Canada, 2 Departement de Biologie Experimentale, Université de Saint Boniface, Winnipeg, Manitoba, Canada, 3 Department of Physiology and Pathophysiology, Rady Faculty of Health Sciences, Max Rady College of Medicine, University of Manitoba, Winnipeg, Manitoba, Canada, 4 Laboratoire d'Immunologie et de Biotechnologie, UR7509-IRMAIC, UFR de Pharmacie, Pôle-Santé, Université de Reims Champagne-Ardenne, Reims, France

* abdel.gounni@umanitoba.ca

## Abstract

### Background

Asthma is marked by chronic airway inflammation, immune dysregulation, and airway remodeling. While TSLP is known to influence allergic diseases like asthma, the role of TSLPR in airway remodeling is not well-defined.

### Methods

Using TSLPR-deficient (TSLPR-/-) mice in a chronic HDM asthma model, we assessed lung function, inflammatory cell infiltration, cytokine levels, and antibody production in serum and lung tissues. Airway remodeling was evaluated by examining mucus production, goblet cell metaplasia, and collagen deposition.

### Results

TSLPR-/- mice showed lower airway resistance, tissue resistance, and tissue elastance compared to wild-type mice after chronic HDM exposure. TSLPR-/- mice also had reduced HDM-specific IgE levels and decreased IL-4, IL-13, and IFN-γ in BALF. However, airway and lung inflammation, including inflammatory cell counts and eosinophil infiltration, were similar between TSLPR-/- and WT mice. Collagen deposition, mucus production, and goblet cell changes were also comparable.

### Conclusion

TSLPR deficiency reduced airway hyperresponsiveness but did not significantly impact lung eosinophil and neutrophil counts or mucus and collagen production in

**Data availability statement:** All figure files are available from the Figshare database (https://figshare.com/articles/figure/_b_TSLPR_deficiency_attenuates_AHR_independently_of_eosinophilia_and_mucus_secretion_in_chronic_HDM_mouse_model_of_allergic_asthma_b_/29602289?file=56386262). This PowerPoint file includes all the figures from the paper. Each graph in every figure is linked to the Prism file, which contains the values used to build the graphs and the analysis.

**Funding:** This work was supported by the Canadian Institutes of Health Research grant (MOP # 115115) to A.S.G. The funders had no role in study design, data collection and analysis, decision to publish, or preparation of the manuscript. There was no additional external funding received for this study.

**Competing interests:** The authors declare no competing interests.

**Abbreviations:** TSLP, Thymic stromal lymphopoietin; TSLPR, TSLP receptor, TSLPR$^{-/-}$, TSLPR knockout, HDM, House Dust Mite; AHR, Airway Hyperresponsiveness; BALF, Bronchoalveolar Lavage Fluid; WT, Wild-Type; ILCs, Innate lymphoid cells; COPD, Chronic Obstructive Pulmonary Disease; OVA, Ovalbumin; DC, Dendritic Cells; Th2, T helper 2; MLN, Mediastinal Lymph Node; STAT3, Signal Transducer and Activator of Transcription 3; JNK, c-Jun N-terminal Kinase; MAPK, Mitogen-Activated Protein Kinase, H&E, Hematoxylin and Eosin; IFN-γ, Interferon Gamma; ASM, Airway Smooth Muscle, HASM, Human Airway Smooth Muscle; SPC, Surfactant Protein C; PAS Periodic acid–Schiff; IgE, Immunoglobulin E; IgG1, Immunoglobulin G1.

a chronic HDM asthma model, highlighting the complex role of TSLP and TSLPR in severe asthma.

## Introduction

Asthma, a prevalent chronic condition impacting around 400 million individuals globally, faces significant management challenges, especially in severe cases, despite the progress made in treatment options [1,2]. It significantly impairs quality of life and imposes a major economic burden, highlighting the need to better understand the mechanisms underlying uncontrolled asthma [3,4].

Chronic severe asthma is characterized by persistent airway inflammation, AHR, and lung structural changes [5,6]. AHR refers to exaggerated airway narrowing in response to nonspecific stimuli and is associated with inflammatory cell infiltration [7]. In patients unresponsive to steroid therapy, AHR is linked to airflow limitation and altered airway mechanics due to remodeling [8]. Repeated exposure to HDM, a common indoor allergen, during early life contributes to irreversible airway damage and remodeling [9,10]. Current therapies inadequately address airway contraction, inflammation, and remodeling in severe asthma. Airway inflammation is a critical multicellular process involving mainly eosinophils, neutrophils, CD4$^+$ T lymphocytes, and mast cells [11,12]. Increasing evidence also implicates structural cells in driving inflammation, remodeling, and airflow obstruction in asthma [13].

Thymic stromal lymphopoietin (TSLP), first identified as a pre-B cell growth factor similar to IL-7, is produced by various immune and structural cells such as epithelial cells, mast cells, airway smooth muscle (ASM) cells, and fibroblasts [14–16,17]. TSLP signals through a heterodimeric receptor composed of TSLPR and IL-7Rα, activating pathways such as STAT3 and STAT5 [18]. It has emerged as a critical factor in allergic diseases, including asthma, atopic dermatitis, and allergic rhinitis [19]. TSLP expression correlates with asthma severity [20] and has also been detected in COPD airways [21,22], suggesting that it is involved in the development of airway inflammation. Lung-specific overexpression of TSLP induces spontaneous asthma-like inflammation in mice [23], while TSLPR-deficient mice show suppressed inflammation in acute models [24]. Ying et al. reported elevated TSLP mRNA in asthmatic bronchial biopsies, correlating with increased Th2 chemokines [25]. In epithelial cells, TLR3 activation or Th2 cytokines (IL-4, IL-13) upregulate TSLP expression, which is suppressed by glucocorticoids [25]. Furthermore, in eosinophilic asthma, IL-4 increases epithelial permeability and upregulates IL-33 and TSLP, amplifying Th2 responses [26]. TSLP-activated dendritic cells express OX40L and drive Th2 differentiation via naïve CD4$^+$ T cells [27,28]. We have shown that TSLP promotes human ASM cell proliferation via STAT3 signaling, implicating it in AHR and remodeling [29].

In this study, we investigated whether the absence of TSLP receptor using TSLPR$^{-/-}$ genetic deletion affects AHR, inflammation, and airway remodelling using an eight-week HDM chronic airway exposure.

## Results

*TSLPR deficient mice display reduced airway resistance, tissue elastance and resistance following chronic HDM challenge.*

Since AHR is a key clinical feature of allergic asthma, we investigated the effect of TSLP receptor deletion on both basal and HDM-induced AHR by evaluating lung function parameters. WT and TSLPR$^{-/-}$ mice received intra-nasal HDM or saline for seven weeks (Fig 1A). Mice were anesthetized, tracheotomized, and challenged with escalating doses of methacholine. TSLPR$^{-/-}$ mice displayed significantly reduced airway resistance compared to their WT counterparts. (Fig 1B), tissue resistance (Fig 1C), and tissue elastance (Fig 1D) following chronic HDM sensitization and challenge. The results indicate that TSLPR plays a role in the regulation of house dust mite-induced AHR.

To characterize the inflammatory response, we examined BALF and lung-infiltrating cells. H&E staining of lung sections revealed airway inflammation (Fig 2A), and cytospins from BALF were used to assess inflammatory cell recruitment (Fig 2B). HDM-exposed TSLPR knockout mice had similar total inflammatory cell numbers in BALF compared to WT mice (Fig 2C and 2D). Additionally, the percentage and total absolute numbers of Siglec F$^{+}$/MHCII-negative eosinophils and CD11b$^{+high}$ GR1$^{high}$ neutrophils, were similar between TSLPR deficient and WT mice (Fig 2E-H, S1 Fig). Therefore, TSLPR deletion did not lower the number of neutrophils and eosinophils in both BALF and lung.

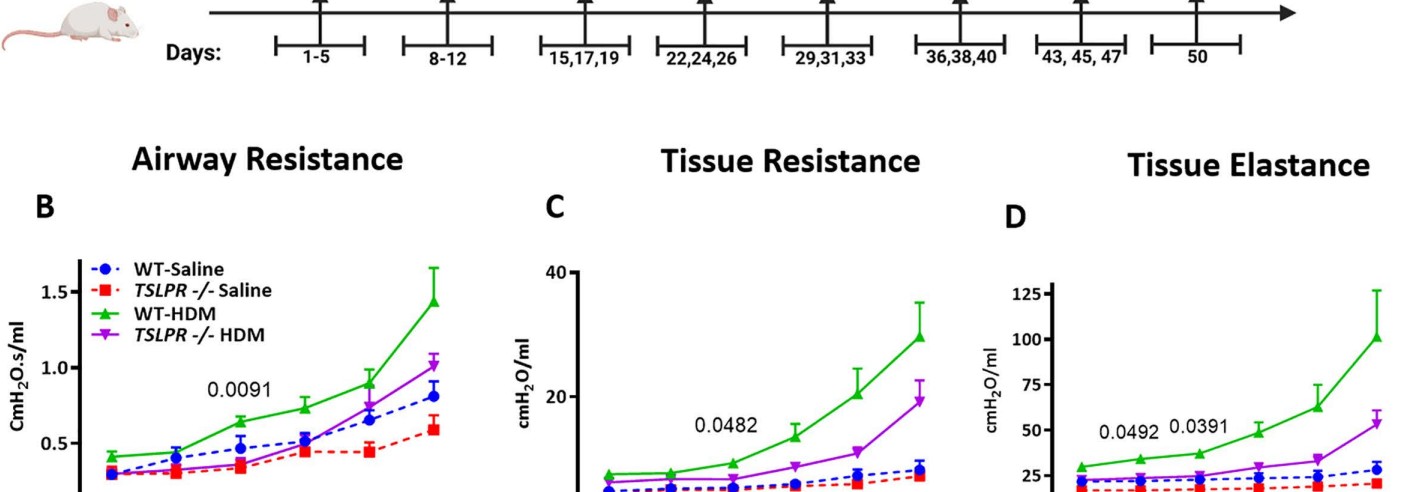

**Fig 1. TSLPR deficient mice show decreased AHR upon HDM exposure. (A)** Chronic model of allergic asthma was developed through intranasal administration of HDM over a 7-week period. TSLPR$^{-/-}$ mice and their WT littermates exposed to HDM were subjected to tracheotomy and subsequent methacholine challenge to assess AHR parameters, including **(B)** airway resistance, **(C)** tissue resistance, and **(D)** tissue elastance. The data presented are from two independent experiments involving age- and sex-matched mice (n = 5-6 mice per group). Statistical significance was determined using repeated measures two-way ANOVA. Results are shown as mean ± SEM of three independent experiments.

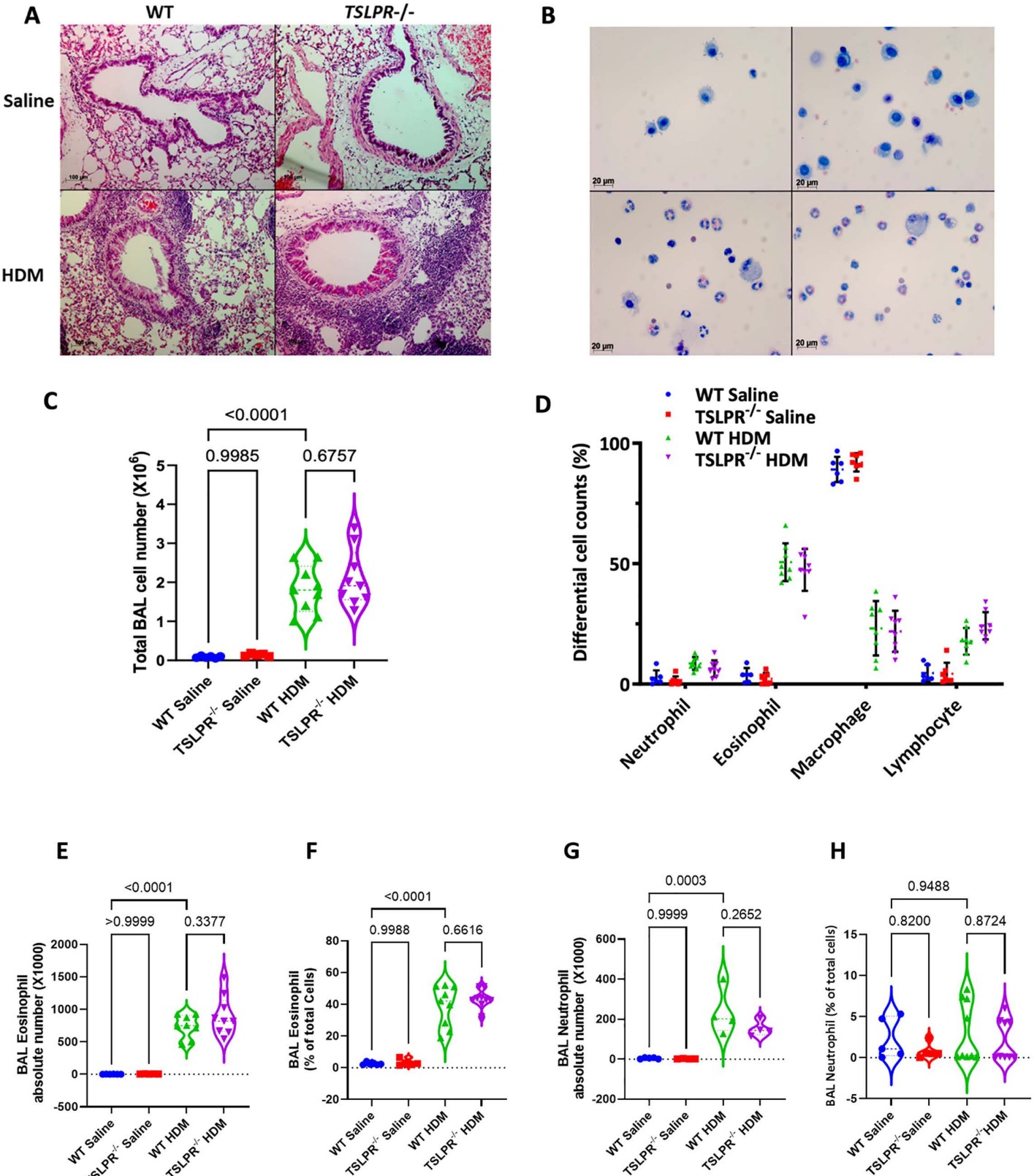

**Fig 2. HDM-induced chronic airway inflammatory cells are not increased in TSLPR deficient mice. (A)** Airway inflammation was examined by conducting H&E staining on lung tissue sections. **(B)** representative cytospin slide of HDM or saline challenged TSLPR-/- and WT mice BALF cells. **(C-D)** Total and differential cell counts were performed using cytospin slides on BAL fluid from both TSLPR-/- and WT mice following saline and HDM challenges. **(E-H)** BAL eosinophils and neutrophils were analyzed by FACS analysis. (n = 4-9 mice per group). (one-way ANOVA). Violin plots display the data distribution with median and interquartile range, while dot plots show mean ± SEM, with each dot representing an individual mouse of three independent experiments.

## The level of inflammatory cytokines is decreased in TSLPR knockout mice in the chronic model of asthma

We assessed cytokine levels involved in allergic asthma pathogenesis in BALF, lung homogenates, and MLNs from WT and TSLPR-/- mice. In BALF, IL-4, IL-13, IFN-γ, and IL-33 were significantly decreased in HDM-exposed TSLPR-/- mice compared to WT (Fig 3A, C, D, F), while IL-5 levels showed a non-significant reduction (Fig 3B). IL-17A and IL-25 was undetectable in BALF. Overall, the absence of TSLPR reduced Type 2 cytokines and IFN-γ in response to HDM.

In lung homogenates, IL-4, IL-5, IFN-γ, and IL-33 were also significantly decreased in TSLPR-/- mice received HDM (Fig 3G, H, J, M), whereas IL-13 and IL-17A levels remained unchanged (Fig 3I, K). Interestingly, the level of IL-25 showed a significant increase in TSLPR-/- mice that were exposed to saline and HDM (Fig 3N). The levels of TSLP did not change in BALF and lung homogenates across groups (Fig 3E, L). In MLNs, IL-4, IL-5, and especially IL-13 were markedly reduced in HDM-exposed TSLPR-/- mice, while IFN-γ was elevated in WT controls. IL-17A levels in MLNs did not differ significantly (S2 Fig). These results indicate that TSLPR deficiency dampens IL-4, IFNγ,and IL-33 inflammatory cytokine production in HDM-induced asthma.

## TSLPR deficient mice displayed reduced HDM-specific IgE production in chronic model of asthma

Given the critical role of IgE in allergic asthma [30], we measured both total and HDM-specific IgE levels in the sera and BALF of TSLPR-/- and WT mice. HDM-specific IgE was significantly lower in TSLPR-/- mice compared to WT in both serum and BALF (Fig 4D, 4H). In contrast, total IgG1, total IgE, and HDM-specific IgG1 levels did not differ significantly between WT and TSLPR-/- under either saline or HDM challenge (Fig 4A-C, 4E-G). These results indicate that TSLPR deficiency reduces HDM-specific IgE production in a chronic asthma model, highlighting its significance in the allergic response mechanism.

## TSLPR-/- mice showed similar production of mucus and collagen compared to wild-type mice

Previous studies have indicated that airway obstruction in chronic asthma is, in part, driven by mucus plugging, a phenomenon characterized by the excessive production of mucus by goblet cells [31,32]. To assess this, we performed PAS staining on lung sections from TSLPR-/- and WT mice. Goblet cell density and mucin gene expression (*Muc5ac, Muc5b*) were similar between the two groups following chronic HDM exposure (Fig 5A–D). We next evaluated airway remodeling by assessing collagen deposition. Peribronchial fibrosis scores and *Col3* mRNA expression were comparable between TSLPR-/- and WT mice (Fig 5E–G). Similarly, total lung hydroxyproline content and Col1 protein levels and showed no significant differences between TSLPR-/- and WT mice (S3 Fig).

Altogether, the deficiency of TSLPR demonstrated similar collagen deposition and goblet cell metaplasia in HDM-exposed mice.

## Discussion

In this chronic HDM-driven asthma model, genetic ablation of TSLPR attenuated airway hyperresponsiveness and reduced Th1 and Th2 cytokine levels and allergen-specific IgE, without altering airway remodeling or total inflammatory cell infiltration. TSLPR-deficient mice showed significantly lower airway resistance and tissue elastance compared to WT, which correlated with reduced IL-4, IL-5, IFNγIL-13 and IL-33 in lung tissue and BALF, and lower HDM-specific serum

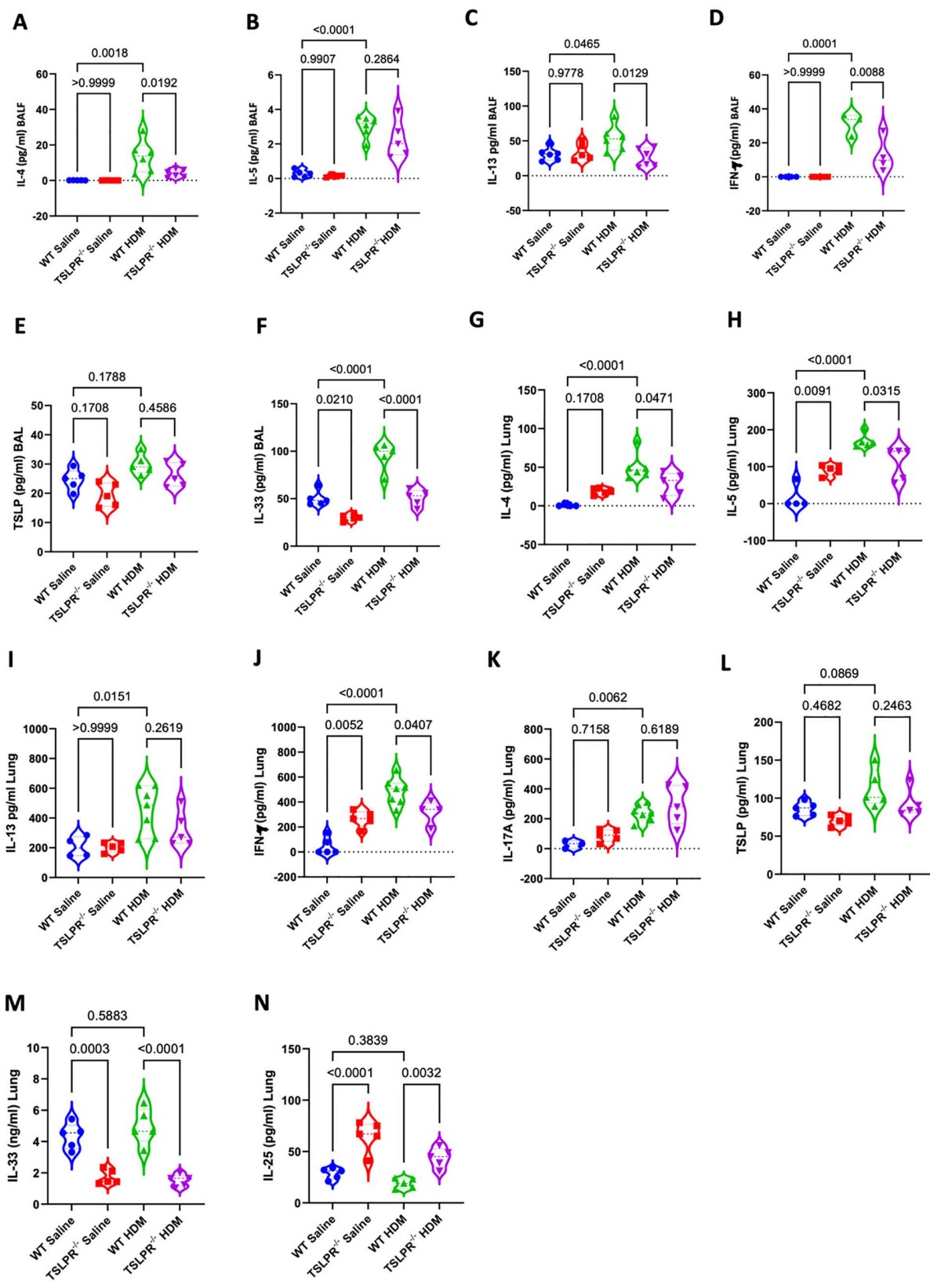

**Fig 3. HDM-induced chronic airway inflammatory cytokines decreased in TSLPR deficient mice. (A-F)** Airway levels of IL-4, IL-5, IL-13, IFN-γ, TSLP and IL-33 were quantified in BALF using ELISA following intranasal exposure to either HDM or saline. **(G-N)** Levels of IL-4, IL-5, IL-13, IFN-γ, IL-17A, TSLP, IL-33, and IL-25 in homogenised lung obtained from TSLPR⁻/⁻ or WT mice were quantified in BALF using ELISA (n = 3-7 mice per group). (one-way ANOVA). Violin plots display the data distribution with median and interquartile range. Results are from three independent experiments.

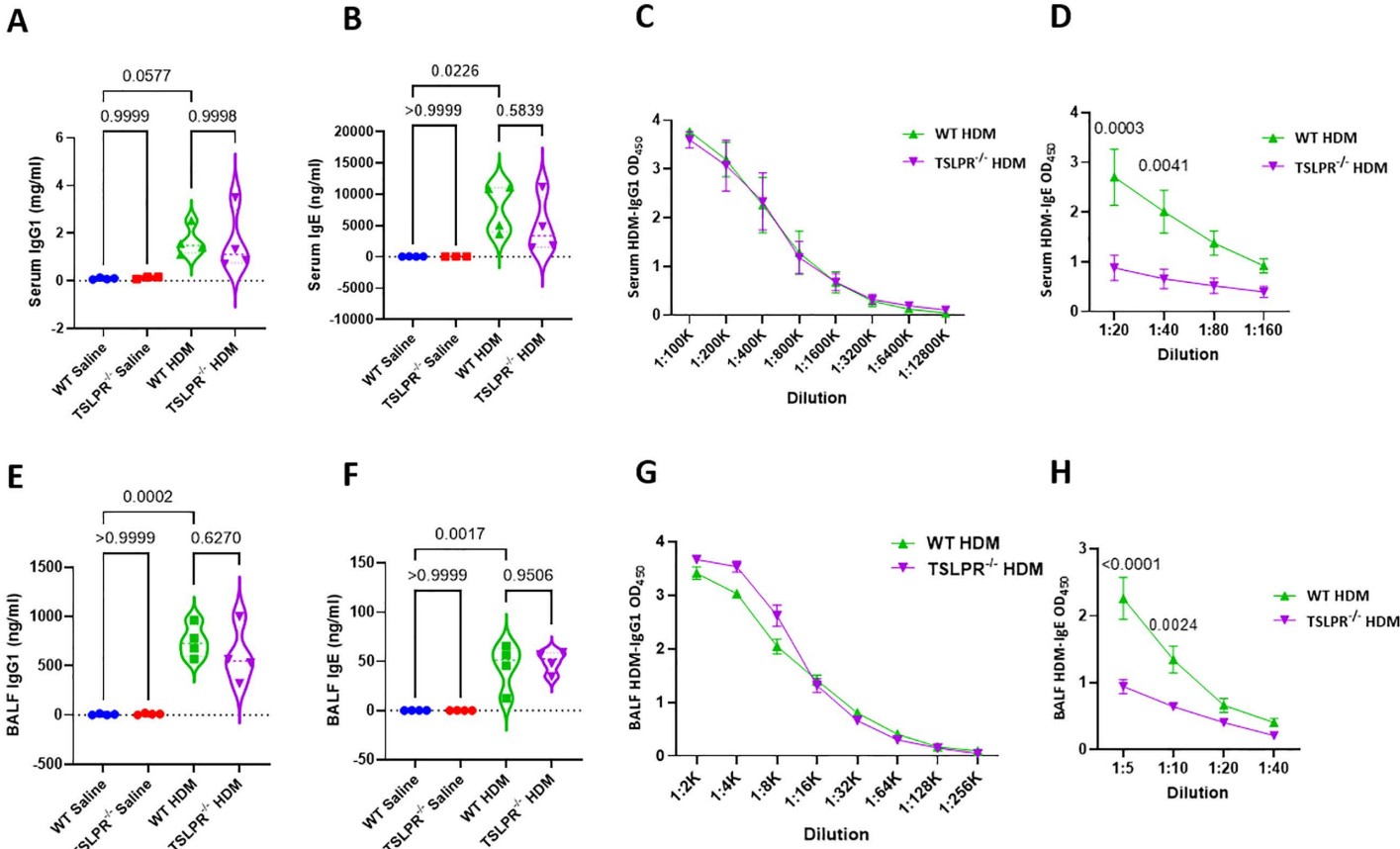

**Fig 4. HDM-specific IgE levels decreased in response to TSLPR ablation in chronic model of asthma mice.** Serum and BALF level of total IgG1 and IgE (**A, B, E, F**) and HDM-specific IgG1 and IgE (**C, D, G, H**) of exposed saline or HDM TSLPR⁻/⁻ and WT mice were measured by ELISA (n = 3-5 mice per group). (one-way ANOVA and repeated measures two-way ANOVA). Violin plots display the data distribution with median and interquartile range, while XY plots show mean ± SEM, with SEM represented by error bars. Results are from three independent experiments.

IgE. However, these changes did not translate into decreased numbers of eosinophils or neutrophils in the BALF and lung tissue, nor did they affect goblet cell hyperplasia, mucus gene expression (*Muc5ac and Muc5b*), or collagen deposition. These results suggest that while TSLP-TSLPR signaling amplifies Th1 and Th2 cytokine levels and AHR, it may be dispensable for maintaining structural airway changes and inflammatory cell accumulation during chronic allergen exposure.

TSLP, an epithelial-derived cytokine, plays a central role in initiating Th2-type immune responses in allergic asthma [33]. In acute models, TSLPR deletion skews the immune response toward Th1, with elevated IL-12, IFN-γ, and IgG2a, and reduced Th2 cytokines and IgE [34]. In line with previous chronic exposure studies, we observed reduced IL-4, IL-5, and IL-13 and lower allergen-specific IgE in TSLPR-deficient mice, confirming the sustained role of TSLP in promoting Th2 immunity during prolonged allergen exposure [35,34]. Interestingly, IFN-γ levels were also reduced in KO mice, and

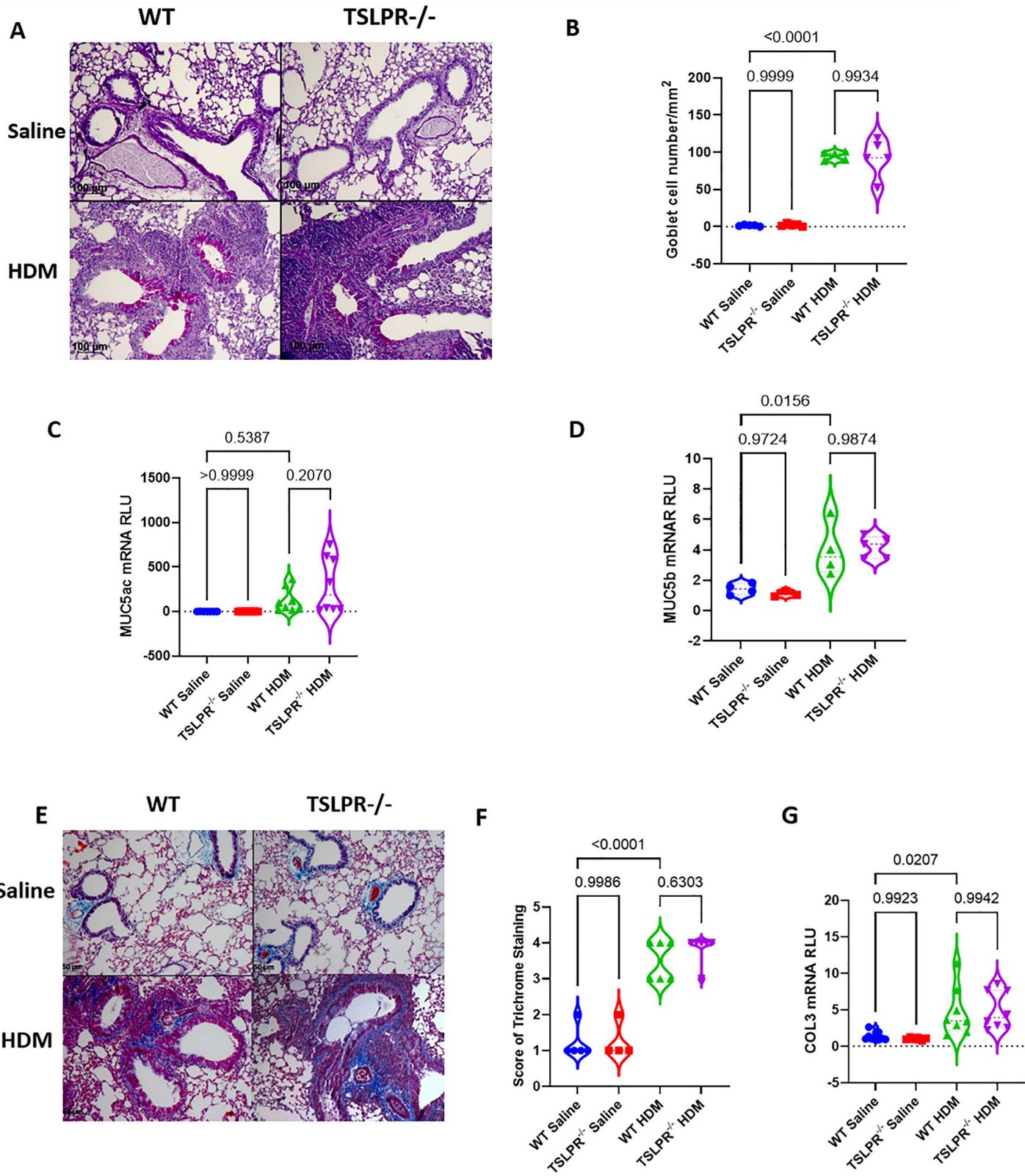

**Fig 5. Mucus and collagen production is similar in TSLPR deficient and WT mice.** Lung tissue sections from TSLPR$^{-/-}$ and WT mice exposed to either saline or HDM were stained with **(A)** PAS to assess mucus hypersecretion. **(B)** the number of goblet cells was reported by mm$^2$,The expression of remodeling-related genes, including **(C)** Muc5ac, **(D)** Muc5b, and Col3 **(G)**, were analyzed using quantitative real-time PCR with specific primers (n = 3-8 mice per group). Collagen were assed by trichrome staining **(E-F)**. Statistical significance was determined using one-way ANOVA. Violin plots display the data distribution with median and interquartile range. Results are from three independent experiments.

IL-17A levels were mildly elevated, albeit not significantly, suggesting a possible shift from Th2 to Th17 responses over time.

AHR is a defining feature of asthma and is largely driven by exaggerated contraction of ASM cells, which may be intrinsically altered or influenced by inflammatory mediators [36–41]. We previously demonstrated that primary human ASM cells express TSLPR and IL-7Rα and respond to TSLP stimulation with increased secretion of IL-6, IL-8/CXCL8, and eotaxin-1/CCL11, driven by MAPKs and STAT3 [40]. We also showed that TSLP promotes ASM migration through STAT3-mediated cytoskeletal changes [29]. Therefore, the reduction in AHR observed in TSLPR-deficient mice might reflects both diminished Th2 inflammation and the absence of direct TSLP signaling to ASM cells. This is consistent with previous work showing that elevated TSLP in lungs correlates with increased AHR in both mouse models and human asthma, and that neutralizing TSLP or downstream cytokines like IL-4 and IL-13 mitigates AHR and airway inflammation [42,43]. Furthermore, we have shown that ASM cells express FcεRI, and IgE activation can stimulate IL-4, IL-5, and IL-13 secretion, while also inducing TSLP expression via Syk and NF-κB signaling [44,45]. In this study, we found lower IgE and reduced Th2 cytokines in TSLPR-deficient mice, disrupting this feed-forward loop between IgE and TSLP, likely contributing to the observed AHR attenuation. However, the the expression of FcεRI and TSLP and its receptor, TSLPR in murine ASM cells is still unknown.

The decreased HDM-specific IgE in TSLPR-deficient mice also underscores the role of TSLP in promoting class-switch recombination via IL-4–dependent pathways [46–48]. TSLP induces OX40L expression in dendritic cells, promoting CD4$^+$ Th2 cell expansion and IL-4 secretion, which in turn facilitates IgE production [49]. In SPC-TSLP mice, TSLP overexpression results in spontaneous Th2 inflammation and elevated IgE levels [50]. Our findings thus support a critical function for TSLP in sustaining the Th2-IgE axis in chronic asthma.

Despite reduced Th2 cytokines and IgE, we observed no significant differences in mucus production, collagen deposition, or expression of remodeling-associated genes in TSLPR-deficient mice. Both TSLPR-deficient and WT mice showed similar levels of Muc5ac, Muc5b, and IL-13 at the endpoint. This may be due to the chronic exposure duration, by five weeks, remodeling features may be well established and no longer reversible by TSLP blockade [51]. Additionally, residual IL-13 expression and persistent eosinophilic inflammation in TSLPR-deficient mice could maintain remodeling via redundant pathways.

The sustained presence of eosinophils and neutrophils, despite reduced Th2 cytokines, suggests compensatory mechanisms. Alarmin cytokines such as IL-25 and IL-33, could have maintained type 2 inflammation. IL-33 is known to enhance eosinophil survival and activation [52], while IL-25 promotes eosinophil recruitment and cytokine production [53]. Moreover, TSLP and IL-33 are thought to form a positive feedback loop in epithelial cells and ILC2s, where TSLP signaling promotes IL-33 release, thereby amplifying type 2 inflammation [54]. It was also shown that TSLPR$^{-/-}$ mice displayed significantly lower Alternaria-induced IL-33 levels compared to WT [54]. Therefore, while our model revealed increased IL-25, this may reflect compensatory epithelial or innate cell responses in the absence of TSLP signaling and attenuation of IL-33-signaling.

Neutrophilic inflammation, often associated with severe or steroid-resistant asthma, is driven by distinct pathways involving IL-8, GM-CSF, and chemokines like CXCL1 and CXCL5 [55,56]. These mediators, likely unaffected by TSLPR deletion, can sustain neutrophil recruitment. The modest rise in IL-17A in TSLPR-deficient mice could also support neutrophilic influx, as IL-17 induces neutrophil-attracting chemokines.

Differences between our findings and earlier studies likely stem from model chronicity. Chronic exposure alters cytokine milieu, cell recruitment kinetics, and remodeling trajectories [57]. Single-time-point measurements, as used here, may obscure earlier differences in inflammation.

In conclusion, our study reveals that TSLP-TSLPR signaling is essential for maintaining chronic Th2 inflammation and AHR but not necessary for inflammatory cell recruitment or structural remodeling in established disease. These findings highlight the complex redundancy of alarmin signaling in chronic asthma and have implications for therapeutic strategies targeting TSLP in severe asthma.

These findings indicate that timing is crucial in capturing the dynamics of immune responses, and further studies should include multiple measurement points to better understand these processes.

## Materials and methods

### Animals

The TSLPR, a chain deficient mouse on Balb/c background, was described previously [58]. Balb/c WT littermates were used as the control group. All experimental procedures were approved by the University of Manitoba Animal Care and Use Committee (protocol number 23–042) and complied with the Canadian Council for Animal Care guidelines and ARRIVE standards. Mice were maintained under specific pathogen-free conditions with free access to food and water. To ensure animal welfare, humane endpoints were clearly defined prior to initiating the study. Animals were monitored at least twice daily for clinical signs of illness or distress, including but not limited to: hunching, labored breathing, reduced mobility, lethargy, piloerection, lack of grooming, and body weight loss exceeding 15% of baseline. Animals meeting any of these criteria were promptly euthanized. Euthanasia was performed under deep anesthesia using isoflurane inhalation, followed by cervical dislocation to ensure death, in accordance with institutional protocols. The time from reaching endpoint criteria to euthanasia did not exceed 12 hours. No animals died unexpectedly prior to meeting euthanasia criteria. All animal handling personnel were trained in humane endpoint recognition and euthanasia procedures. No analgesics were required, as the study involved non-surgical procedures and no anticipated pain beyond transient immune responses. All efforts were made to minimize discomfort and distress.

### House dust mite exposure model

Lyophilized HDM protein extract was obtained from Greer Laboratories (*D. pteronyssinus,* Lot #259585, LPS: 615 EU/vial, Protein: 5.35 mg/vial, Dry weight: 23.73 mg/vial), Lenoir, NC. Working concentration (25 mg per mouse) was freshly prepared. Female and male TSLPR$^{-/-}$ 6–8-week-old mice were subjected to intranasal administration of HDM (25ug) under gaseous anesthesia for 5 days per week during the first two weeks. HDM was continuously administered via intranasal route every three days for the next 5 weeks, with 2 days interval between those weeks [59]. The final HDM administration was performed on day 47 and the mice were sacrificed on day to measure the outcomes. Control WT groups were exposed to the sterile saline in all experiments.

### Methacholine challenge test

Pulmonary mechanics following intravenous injections of increasing concentrations of methacoline (Sigma-Aldrich, Sweden) were assessed in a flexiVent apparatus (Scireq, Montreal, Canada). Mice were anesthetized i.p. with penthobarbital (70–90 mg·kg-1 body weight; Apoteket Produktion & Laboratorier AB, Stockholm, Sweden), tracheotomized with a blunted 18-gauge cannula and ventilated (flexiVent, SCIREQ, Montreal, Canada). Using the forced-oscillation technique, ventilation was performed at 2.5 Hz in a quasi-sinusoidal fashion to generate a sinusoidal pressure waveform during lung inflation. Utilizing the forced-oscillation technique, pulmonary resistance was determined assuming a single-compartment linear model. Resistance in conducting airways and tissue resistance and tissue elastance were evaluated by assuming a constant-phase model. The flexiVent software calculates a coefficient of determination (COD) to establish the goodness of fit of this model to the data. Any data point with a COD < 0.9 is automatically rejected.

## Collection and processing *of* bronchoalveolar lavage fluid

BALF was collected from tracheally canulated *TSLPR*[-/-] or WT mice with two instillations of 1 ml of sterile saline containing 0.1 mM EDTA. After lysis of RBC with ACK buffer, total BALF cells were counted and cytospins were prepared. Then, cytospins were fixed and stained with H&E to perform a differential inflammatory cell count. BALF supernatants were stored at −80°C for future analysis [60].

## FACS analysis *of* airway inflammatory cells

After RBC lysis and Fc blocking, cells were stained with a cocktail containing the following anti-mouse antibodies: Gr-1-Pacific blue (Clone: 30-F11, eBioscience), Siglec F-PE (Clone: E50-2440, BD Biosciences), CD11b-APC (Clone: M1/70, eBioscience), CD11c-PE (Clone: N418, eBioscience), MHCII-FITC (Clone: 17A2, eBioscience), CD45-eFluor® 450 (Clone: RA3-6B2, eBioscience), F4/80-PE (Clone:, eBioscience). Then, the samples were acquired using a BD FACSCanto-II flow cytometer and analyzed using FlowJo software [60].

## RNA isolation and real-time PCR

The right middle lobe of the lungs was homogenized in Trizol® (Life Technologies, Burlington, ON), followed by RNA isolation and cDNA synthesis as per the manufacturer's instructions. The expression of Col3 and Muc5ac was then analyzed using specific murine primers.

## Lung histology

Lower left lobe of the lung was dissected, inflated, and fixed in formalin overnight followed by embedding in paraffin. Basal and HDM-induced airway inflammation, mucus production, and collagen deposition in lung tissue sections obtained from *TSLPR*[-/-] versus WT mice was studied by performing H&E, Periodic Acid-Schiff (PAS) and trichome, respectively. Morphometric analysis of PAS-stained slides was utilized to quantify the mucus overproduction in the airways as we described previously [59].

## Hydroxyproline assay

The procedure involved the homogenization of lung tissue, followed by drying it until a consistent weight was achieved. Subsequently, the tissue was acidified using 6 M hydrochloric acid (HCl) and subjected to hydrolysis through heating at 120°C for a duration of 24 hours. Hydroxyproline measurements were then conducted according to previously established methods and adjusted based on the dry weight of the lung tissue.

## Measurement *of* cytokines and immunoglobulins

ELISA of mouse BALF for IL-4, IL-5, IL-13, IL-17A, and IFN-γ was performed according to the manufacturer's instructions. ELISA plates were read with SpectraMax plate reader and analyzed with SoftMax Pro software (Molecular Devices). All cytokine ELISA kits were from BioLegend (San Diego, CA) except IL-13 and IL-33 which were from eBioscience (San Diego, CA) and R&D Systems (Minneapolis, MN), respectively. Serum was obtained from saline- and HDM-exposed WT and *TSLPR*[-/-] mice 48h after the last allergen challenge. Total and HDM-specific IgE and IgG1 levels were quantified using commercial ELISA kits according to manufacturer's instructions (Southern Biotech Birmingham, Al) as we described previously [59].

## Intracellular staining *of* cytokines

Lung draining mediastinal lymph nodes (MLN) were collected from HDM-exposed mice followed by preparing single cell suspension using a cell strainer. The cells were re-suspended at a concentration of 4x10⁶ cells/ml in DMEM supplemented

with 10% FBS, 2 mM L-glutamine, 100 U/ml penicillin, 100 U/ml streptomycin, and 5x10⁻⁵ M 2-ME, plated in 24-well tissue culture plates. Then, MLN cells were incubated with a freshly prepared cocktail containing 50 ng/ml PMA, 500 ng/ml ionomycin and 10 µg/ml brefeldin A (BFA) (Sigma-Aldrich, Oakville, Ontario, Canada) for 4h at 37°C and 5% $CO_2$. Extracellular staining was performed by using anti-mouse CD3 e-Fluor® 450 (Clone: 17A2) and CD4-FITC both from eBioscience. For staining cytokines, fixed and surface-stained MLN cells were permeabilized with 0.1% saponin in flow cytometry buffer and then stained with specific anti-mouse IFN-γ-PE (Clone: XMG1.2), IL-4-APC (Clone: 11B11) and IL-17A-PE (Clone: eBio17B7) all from eBioscence. Samples were acquired on a FACSCanto II and analyzed using FlowJo software as we described previously [59].

### Ex-vivo mediastinal lymph node cell culture *for* recall response

WT and TSLPR⁻/⁻ mice were challenged with HDM and then MLN were collected. Single cell suspension of MLN cells was prepared by grinding them in a 40µm strainer (BDfalcon). RBCs were lysed by incubating spleen cells in NH4Cl solution. Cell suspensions were then centrifuged and resuspended in complete cell culture media (RPMI supplemented with 10% fetal bovine serum and 20 µM 2-mercaptoethanol). Cells were cultured in complete RPMI at 2X10⁶ cells/ml in a 48 well plate at 37°C in a 5% $CO_2$ atmosphere, treated with HDM (10ug/ml) for 72 hrs and supernatant was then collected after 72 hrs and stored at −80°C freezer for ELISA.

### Western blotting

Whole-cell protein lysate was collected using RIPA buffer in the presence of protease inhibitors and PMSF. After protein concentration was determined, the lysate was denatured in 2 × Laemmli Sample Buffer, separated by electrophoresis, and transferred to nitrocellulose membrane. Membranes were blocked with 5% non-fat dry milk in TRIS buffer containing 0.1% Tween-20 (TBST) for 1 hr followed by incubation with primary antibody overnight at 4°C. The membranes were then washed with TBST followed by application of appropriate HRP-conjugated secondary antibody for 1 hr at room temperature and bands were revealed with electrochemiluminescence reagents. GAPDH were used as loading control. Densitometric analysis was performed by using AlphaEase FC software (Alpha Innotech, San Leandro, Calif).

### Statistics

GraphPad Prism 9.0 software was used for statistical analysis and values were presented as the mean±SEM. Depending on the number of groups and treatments, data were analyzed by one-way or repeated measures two-way ANOVA, followed by the Bonferroni's multiple comparisons post-hoc test. P-values are reported on the figures.

### Supporting information

**S1 Fig. Flow Cytometry Gating Strategy for BALF Eosinophils and Neutrophils.** (A) Eosinophils were identified by gating on Siglec-F⁺ and MHCII⁻ cells. Initial gating steps included selecting live, single cells to exclude debris and doublets, followed by gating on Siglec-F and MHCII expression to isolate the eosinophil population. The representative sample shown is from the HDM-exposed group, selected due to the higher number of cells recovered in BALF, allowing for clearer visualization of the gating strategy. (B) Neutrophils were identified by gating on CD11b⁺ and GR1⁺ cells, with eosinophils excluded.
(PDF)

**S2 Fig. Recall cytokine response in mediastinal lymph node is decreased in TSLPR deficient mice.** Single-cell suspension of mediastinal lymph node from the mice exposed to HDM or saline was prepared and stimulated with medium or HDM *in vitro.* The levels of IL-4, IL-5, IL-13, IL-17A, and IFN-γ were measured in the supernatant of the cultured cells after

72 hours by ELISA (n = 3–8 per group). *p < 0.05, **p < 0.01, and ***p < 0.001 (one-way ANOVA). Violin plots display the data distribution with median and interquartile range. Results are from three independent experiments.
(PDF)

**S3 Fig. The production of Collagen1 was similar and did not show significant difference between TSLPR$^{-/-}$ and WT mice.** (A) The lung tissue lysates were subjected to immunoblot analysis with specific antibodies against collagen 1 and glyceraldehyde 3-phosphate dehydrogenase (GAPDH). (B) Expression level of collagen 1 was quantifies by densitometry and normalized with GAPDH. (C) Whole lung hydroxyproline levels in mice. Data are represented as the mean±SEM. Means with different superscript letters are significantly different from one another (*P < 0.05). Results are shown as mean±SEM of three independent experiments.
(PDF)

## Author contributions

**Conceptualization:** Abdelilah S. Gounni.

**Data curation:** Latifa Koussih, Sina Taefehshokr, Lianyu Shan, Sujata Basu.

**Formal analysis:** Latifa Koussih, Sina Taefehshokr, Lianyu Shan.

**Investigation:** Abdelilah S. Gounni.

**Methodology:** Latifa Koussih, Sina Taefehshokr, Lianyu Shan.

**Project administration:** Abdelilah S. Gounni.

**Software:** Latifa Koussih, Sujata Basu.

**Supervision:** Andrew Halayko, Bouchaib Lamkhioued, Abdelilah S. Gounni.

**Validation:** Andrew Halayko, Bouchaib Lamkhioued.

**Visualization:** Sina Taefehshokr, Lianyu Shan.

**Writing – original draft:** Sina Taefehshokr.

**Writing – review & editing:** Sina Taefehshokr.

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
