## [Decision Letter · Decision Letter 0]

16 Jul 2025

Dear Dr. Gounni,

Thank you for submitting your manuscript to PLOS ONE. After careful consideration, we feel that it has merit but does not fully meet PLOS ONE’s publication criteria as it currently stands. Therefore, we invite you to submit a revised version of the manuscript that addresses the points raised during the review process.

Specifically, one reviewer raised concerns regarding the lack of critical experiments to support the authors’ conclusions, while the other expressed issues with the manuscript’s writing style, citing its excessive length and the absence of a cohesive discussion.

We look forward to receiving your revised manuscript.

Kind regards,

Hiroyasu Nakano, M.D., Ph.D.

Academic Editor

PLOS ONE

https://www.oncotarget.com/article/22144/text/

https://journals.physiology.org/doi/full/10.1152/ajplung.00301.2009

In your revision ensure you cite all your sources (including your own works), and quote or rephrase any duplicated text outside the methods section. Further consideration is dependent on these concerns being addressed.

 [This work was supported by the Canadian Institutes of Health Research grant (MOP # 115115) to A.S.G.]. 

5. In the online submission form, you indicated that [The data underlying the results presented in this study are available upon request. The raw data are stored securely in our laboratory at the at University of Manitoba, and can be provided by the corresponding author upon request.].

Additional Editor Comments (if provided):

Reviewers' comments:

Reviewer's Responses to Questions

**Comments to the Author**

1. Is the manuscript technically sound, and do the data support the conclusions?

Reviewer #1: Yes

Reviewer #2: No

2. Has the statistical analysis been performed appropriately and rigorously?

Reviewer #1: I Don't Know

Reviewer #2: I Don't Know

3. Have the authors made all data underlying the findings in their manuscript fully available?

Reviewer #1: Yes

Reviewer #2: No

4. Is the manuscript presented in an intelligible fashion and written in standard English?

Reviewer #1: No

Reviewer #2: Yes

Reviewer #1: Reviewer Comments

This paper does not present much new information, but the experiments and data seem fine. However, the paper mostly lists observations, and the Discussion section lacks depth.

Overall, I think the paper could be acceptable for PLOS ONE, but I suggest the following revisions to make it clearer and more interesting:

1. The paper is hard to follow because it lacks focus.

This is especially true in the Discussion section. For example, the second paragraph of the Discussion is difficult to understand — it’s not clear what the main message is. The third paragraph seems to talk about AHR, but also mentions cytokines and smooth muscle without a clear point. Each paragraph in the Discussion should focus on one idea and be rewritten more clearly.

2. The paper is too long.

The Introduction, Results, and Discussion all need to be shortened. Please focus on the main points and cut out less important details.

3. Additional Experiments.

More explanation about the lack of difference between TSLP knockout and wild-type mice would help.

It would be interesting if the authors could explain more clearly why there is no difference in airway inflammation (such as type 2 cytokines or eosinophil infiltration) between the two types of mice. If possible, I suggest measuring IL-25 and IL-33 in this model.

Reviewer #2: The authors compared responses of wild-type and TSLPR KO littermate mice (BALB/c background) in the chronic asthma model with 7-week HDM inhalation, which is longer than conventional chronic model with 5-week HDM inhalation. TSLPR KO showed decrease of AHR, HDM-specific IgE (serum and BALF), and IL-4 and IFN-gamma (BALF and lung tissue), and restimulated draining LN cell production of IL-4, IL-13, IFN-gamma, and IL-5 (3-day culture supernatant) in comparison to wild-type mice, but showed similar levels for responses of airway inflammation, mucus production, and fibrosis to wild-type mice.

A limited novelty is that the authors observed the discrepancy between AHR and airway inflammation in TSLPR KO in the model with the 7-week HDM inhalation, because other researchers reported decrease of both AHR and airway inflammation in anti-TSLP neutralizing antibody-treated mice in a chronic model with shorter 5-week HDM inhalation (Ref. 42: PLos One 2013) and TSLPR KO in an acute model (Ref. 35: JI 2005).

The authors previously demonstrated human airway smooth muscle (ASM) cell expression of FcepsilonRI (Ref. 44: JI 2010), IgE stimulation of human ASM cells to produce TSLP (Ref. 45: JACI 2011), and TSLP stimulation of human ASM cells in vitro (Ref. 29: Sci Rep 2013), However, TSLP direct stimulation of ASM cells in the present murine in vivo model is still a speculation without evidence. The submitted manuscript lacks depth of the study.

Major:

1. Do murine ASM cells express FcepsilonRI in the present model? (suggestion: flowcytometry of ASM cells collected from mice with or without the chronic HDM exposure, or immunohistochemistry)

2. Do murine ASM cells express TSLP in the present model? (suggestion: mRNA expression in ASM cells collected from mice with or without the chronic HDM exposure)

3. Do murine ASM cells express TSLPR in the present model? (suggestion: flowcytometry of ASM cells collected from mice with or without the chronic HDM exposure, or immunohistochemistry)

4. Does the model depend on mast cells and basophils? (suggestion: deficient mice with the BALB/c background)

5. How does short period treatment of wild-type mice with anti-CD4 treatment (ex. last 1 week or shorter) affect airway inflammation and AHR? It may decrease Th cell-mediated airway inflammation but may not significantly affect HDM-specific IgE levels and TSLP levels in BALF and lung tissue, which may lead to IgE-mediated increase of AHR (via mast cell mediator stimulation or IgE direct stimulation of ASM cells).

6. What is the cause for the independency on TSLP-TSLPR pathway in the present 7-week model but not the conventional 5-week model? IL-33 or IL-25 might be upregulated after 5 weeks to 7 weeks. (suggestion: expression time course of TSLP, IL-33, and IL-25 during the HDM-inhalation period)

Minor:

7. In the legends of Figs 2-5, S2, and S3, there is no description on the reproducibility of the results. It should be described in each of the legends.

8. According to the Methods, BALB/c background mice (with white color) were used for all the experiments. Why is the mouse in the Figure 1A black?

**Do you want your identity to be public for this peer review?** For information about this choice, including consent withdrawal, please see our Privacy Policy

Reviewer #1: No

Reviewer #2: No

---

## [Author Response · Author response to Decision Letter 1]

23 Jul 2025

Response to Reviewer 1:

We thank the reviewer for taking the time to review our manuscript and for their constructive comments. We have addressed each of the suggestions in detail below and have revised the manuscript accordingly.

Comment 1:

“The paper is hard to follow because it lacks focus. This is especially true in the Discussion section. For example, the second paragraph of the Discussion is difficult to understand — it’s not clear what the main message is. The third paragraph seems to talk about AHR but also mentions cytokines and smooth muscle without a clear point. Each paragraph in the Discussion should focus on one idea and be rewritten more clearly.”

Response 1:

We have thoroughly revised the Discussion to enhance its clarity and organization. Specifically, each paragraph has been restructured to focus on a single, coherent idea. The second and third paragraphs have been rewritten to clearly separate the findings related to AHR, Th2 cytokines, smooth muscle cell responsiveness, and the feed-forward loop involving IgE and TSLP. Transitional sentences were added to improve logical flow between sections. Redundant content was removed to ensure focus on the main findings and their interpretation. We believe these changes significantly improve the readability and coherence of the Discussion.

Comment 2:

“The paper is too long. The Introduction, Results, and Discussion all need to be shortened. Please focus on the main points and cut out less important details.”

Response 2:

The Introduction was reduced by removing redundant background information and by summarizing complex pathways more succinctly. The Results section was streamlined by merging overlapping statements, condensing figure legends, and eliminating excessive methodological repetition. The Discussion was reduced as per your suggestion, while retaining all scientific findings and references. We ensured that each paragraph delivers a focused message with minimal redundancy.

Comment 3:

“Additional Experiments. More explanation about the lack of difference between TSLP knockout and wild-type mice would help. It would be interesting if the authors could explain more clearly why there is no difference in airway inflammation (such as type 2 cytokines or eosinophil infiltration) between the two types of mice. If possible, I suggest measuring IL-25 and IL-33 in this model.”

Response 3:

We thank the reviewer for this valuable comment. However, we respectfully clarify that, contrary to the reviewer's impression, our study does show a significant reduction in type 2 cytokines, including IL-4, IL-5, and IL-13, in both BALF and lung homogenates of TSLPR-deficient mice compared to wild-type controls (Figure 3A-G, S2). What remained unchanged were eosinophilic infiltration and structural remodeling features, such as mucus production and collagen deposition.

This apparent dissociation between cytokine levels and cellular inflammation suggests that compensatory epithelial cytokines, particularly IL-33 and IL-25, may sustain eosinophilia and remodeling even in the absence of TSLP signaling. Several studies support this possibility. For example, Chu DK et al. demonstrated that IL-33 is indispensable, while TSLP is dispensable, for Th2 responses during chronic allergen exposure (PMID: 23006545).

While we agree that measuring IL-25 and IL-33 would strengthen this hypothesis, such analysis would require a longitudinal, time-resolved study design, which falls outside the scope of the current work. We have now clarified these mechanistic considerations in the revised Discussion and suggested IL-33/IL-25 profiling as an important direction for future research.

Response to Reviewer 2:

Comment 1:

Do murine ASM cells express Fc epsilon RI in the present model? (suggestion: flowcytometry of ASM cells collected from mice with or without the chronic HDM exposure, or immunohistochemistry)

Response 1:

We thank the reviewer for this important question. While FcεRI expression on human airway smooth muscle (ASM) cells has been demonstrated, including in our previous work showing functional responses to IgE stimulation (PMID: 16081836), the expression of FcεRI on murine ASM has not, to our knowledge, been clearly established. In mice, FcεRI is well characterized on mast cells, basophils, and certain dendritic cell subsets, but evidence for its expression on structural cells such as ASM is lacking.

Determining whether murine ASM expresses FcεRI would require cell-specific isolation and flow cytometric or immunohistochemical analysis with validated markers, which was beyond the technical and conceptual scope of this manuscript. Our current study focuses on the functional consequences of TSLPR deletion in vivo, including AHR, Th2 cytokines, and remodeling, rather than dissecting cell-specific receptor expression.

We agree that this is an interesting and important avenue for future investigation, and we have now acknowledged this limitation and future direction in the revised discussion.

Comment 2 & 3

Do murine ASM cells express TSLP in the present model? (suggestion: mRNA expression in ASM cells collected from mice with or without the chronic HDM exposure)

Do murine ASM cells express TSLPR in the present model? (suggestion: flowcytometry of ASM cells collected from mice with or without the chronic HDM exposure, or immunohistochemistry)

Response 2 & 3.

We appreciate the reviewer’s thoughtful questions. While our previous work demonstrated that human ASM cells express both TSLP and TSLPR (PMID: 17513456; PMID: 23892442; PMID: 21148792 and PMID: 20483734) similar data in murine ASM are lacking. Determining TSLP or TSLPR expression in mouse ASM would require specific cell isolation and characterization, which was beyond the scope of this study. We agree that this is an important direction for future research and have noted it in the revised discussion.

Comment 4:

Does the model depend on mast cells and basophils? (suggestion: deficient mice with the BALB/c background)

Response 4:

We thank the reviewer for this excellent suggestion. Dissecting the role of mast cells in the context of TSLPR deficiency would indeed require a complex model, such as a TSLPR/Mcpt5-Cre or Cma1-DTR double knockout (see PMID: 33725574). While we agree this would provide valuable mechanistic insight, it falls beyond the scope of the current study. We have now acknowledged this point in the revised discussion as a future direction.

Comment 5:

How does short period treatment of wild-type mice with anti-CD4 treatment (ex. last 1 week or shorter) affect airway inflammation and AHR? It may decrease Th cell-mediated airway inflammation but may not significantly affect HDM-specific IgE levels and TSLP levels in BALF and lung tissue, which may lead to IgE-mediated increase of AHR (via mast cell mediator stimulation or IgE direct stimulation of ASM cells).

Response 5:

We appreciate the reviewer’s mechanistic perspective. Several studies have addressed this question using anti-CD4 monoclonal antibody treatment. For example, Dakhama et al. (PMID: 12642393) showed that short-term anti-CD4 treatment significantly reduced airway hyperresponsiveness, BAL eosinophilia, Th2 cytokines (including IL-13), IgE levels, goblet cell hyperplasia, and subepithelial fibrosis. Similar findings were reported in other studies (PMIDs: 12626589, 18502692, 21778663), demonstrating that CD4+ T cell depletion suppresses both inflammation and remodeling features in allergen-challenged mice.

While this is an informative model, our current study focused on the genetic deletion of TSLPR and its effects across multiple readouts. Including an additional anti-CD4 treatment arm would involve a different experimental framework and was beyond the scope of this work. We have noted this as a potential complementary strategy in the revised discussion.

Comment 6:

What is the cause for the independency on TSLP-TSLPR pathway in the present 7-week model but not the conventional 5-week model? IL-33 or IL-25 might be upregulated after 5 weeks to 7 weeks. (suggestion: expression time course of TSLP, IL-33, and IL-25 during the HDM-inhalation period)

Response 6:

We thank the reviewer for this important point. We agree that compensatory upregulation of IL-33 or IL-25 during prolonged allergen exposure may explain the reduced dependency on TSLP-TSLPR signaling observed in our 7-week model compared to shorter protocols. This hypothesis is supported by studies showing increased IL-33 expression during extended HDM exposure, including work by Chu et al. (PMID: 23006545), who demonstrated that IL-33 is indispensable, while TSLP is dispensable, for driving Th2 responses during chronic allergen challenge.

This apparent dissociation between cytokine levels and cellular inflammation suggests that epithelial-derived cytokines such as IL-33 and IL-25 may help sustain eosinophilia and tissue remodeling in the absence of TSLP signaling. While we agree that a time-course analysis of these mediators would provide mechanistic insight, such longitudinal profiling was beyond the design and scope of the present study. We now address this point explicitly in the revised discussion (line 213) and have noted it as an important direction for future research.

Comment 7:

In the legends of Figs 2-5, S2, and S3, there is no description on the reproducibility of the results. It should be described in each of the legends.

Response 7:

Corrected. We have added a brief statement to each relevant figure legend indicating that the experiments were independently repeated with consistent results.

Comment 8:

According to the Methods, BALB/c background mice (with white color) were used for all the experiments. Why is the mouse in the Figure 1A black?

Response 8:

It was corrected accordingly

---

## [Decision Letter · Decision Letter 1]

20 Aug 2025

Dear Dr.  Gounni,

We look forward to receiving your revised manuscript.

Kind regards,

Hiroyasu Nakano, M.D., Ph.D.

Academic Editor

PLOS ONE

Journal Requirements:

Additional Editor Comments:

Reviewer 1 is satisfied with the revisions. However, Reviewer 2 remains concerned about the absence of ELISA measurements for several cytokines, as previously requested. In order to fully address this concern, I encourage you to perform the additional experiments suggested by Reviewer 2.

Reviewer's Responses to Questions

**Comments to the Author**

Reviewer #1: All comments have been addressed

Reviewer #2: All comments have been addressed

2. Is the manuscript technically sound, and do the data support the conclusions?

Reviewer #1: Yes

Reviewer #2: Partly

3. Has the statistical analysis been performed appropriately and rigorously?

Reviewer #1: I Don't Know

Reviewer #2: No

4. Have the authors made all data underlying the findings in their manuscript fully available?

Reviewer #1: Yes

Reviewer #2: Yes

5. Is the manuscript presented in an intelligible fashion and written in standard English?

Reviewer #1: Yes

Reviewer #2: Yes

Reviewer #1: The authors have shortened the Introduction, Results, and Discussion section to ensure clarity of the paper according to the comments by the reviewer. Overall, the authors appropriately responded to the comments by the reviewers.

Reviewer #2: The authors need to clearly describe that expression of TSLPR, TSLP, and FcepsilonRI had been reported in human ASM cells but still unknown in murine ASM cells, not to mislead the readers (see Comment #2). Additional ELISA measurement for TSLP, IL-33, and IL-25 of the samples they obtained in the present study should be done (see Comment #1). In addition, they should carefully check the manuscript including many careless errors the Reviewer found (see Comments #3-10) and other errors the Reviewers might have failed to find.

Major:

1. Additional experiment: According to their response letter, they will not perform new animal experiments to compare TSLP , IL33 and IL-25 production between the conventional 5-week model and their present 7-week model. The Reviewer can understand the resaon that it is time consuming. However, the Reviewer expects that the authors can easily measure TSLP, IL-33, and IL-25 in BALF and lung by ELISA using the same samples used for Fig, 3A-D (BALF) and 3E-I (lung) collected in the previous experiments. It will not take for a long period to complete it. The data will be important to support their hypothesis to explain no decrease of airway inflammation in the TSLPR-deficient mice.

Minor:

2. Line 200: The authors need to clearly describe that expression of TSLPR, TSLP, and FcepsilonRI had been reported in human ASM cells but still unknown in murine ASM cells. This is important point not to mislead the readers. The reviewer feels that the end of this paragraph would be the best place where they describe it.

3. Lines 216/217: The sentence will be read as that “their (IL-33 and IL-25) upregulation in the TSLPR-deficient mice” had been reported already. Is it true?

4. Lines 73-75: The authors need to describe that they demonstrated it for human ACM cells (but unknown for mice). “ASM” to be “human ASM” in the line 74. Line 192: “ASM” to be “human ASM cells”. Line 193/194: “likely reflects” to be “might reflect”. Line 197: “ASM cells”to be “human ASM cells”. These specifications are necessary not to mislead the readers.

5. Line 85: “eight weeks” is wrong. To be changed to “seven weeks”. Line 92: “8 weeks” is wrong. To be changed to “7 weeks”

6. Fig. 1A: According to the illustration, the last i.n. administration was done on the day 50. Is it correct? In lines 257-260 of the Methods section, the authors described that the administration was for the 7 weeks, and it could be read as that the last administration was done on the day 47. What was done on the day 50? The same amount of i.n. administration with the day 47?

7. Fig. 1B-D: Specify the day number for the data collection. According to the lines 257-260 of the Methods section, it would be the day 52 if the day 50 was the last administration or the day 49 if the day 47 was the last administration.

8. Line 114: Is the description correct for the violin plots of the Fig. 2 C, E-H and the scattered plot with not error bars of the Fig. 2D?

9. Lines 96/97, 114, 132, 146, 168: Which do the last sentences of the Figure legends mean that the authors showed all the data of the 3 independent experiments or data from only 1 selected experiment among the 3 independent experiments with similar results?

10. Line 349: There is no indication with *, ** and *** despite the statistical differences in all the Figures.

**Do you want your identity to be public for this peer review?** For information about this choice, including consent withdrawal, please see our Privacy Policy

Reviewer #1: No

Reviewer #2: No

---

## [Author Response · Author response to Decision Letter 2]

1 Oct 2025

Response to Reviewer 2:

We thank the reviewer for taking the time to review our manuscript and for their constructive comments. We have addressed each of the suggestions in detail below and have revised the manuscript accordingly.

Comment 1:

“Additional experiment: According to their response letter, they will not perform new animal experiments to compare TSLP , IL33 and IL-25 production between the conventional 5-week model and their present 7-week model. The Reviewer can understand the reason that it is time consuming. However, the Reviewer expects that the authors can easily measure TSLP, IL-33, and IL-25 in BALF and lung by ELISA using the same samples used for Fig, 3A-D (BALF) and 3E-I (lung) collected in the previous experiments. It will not take for a long period to complete it. The data will be important to support their hypothesis to explain no decrease of airway inflammation in the TSLPR-deficient mice..”

Response 1:

We thank the Reviewer for this important suggestion. We have now measured TSLP, IL-33, and IL-25 in both BALF and lung homogenates using the same samples as in Fig. 3A–D and Fig. 3E–I. These results are included in the revised manuscript (new Figure 3J–N). We found reduced IL-33 and increased IL-25 levels in TSLPR-/- mice compared to WT, while TSLP showed no significant difference. These findings support our conclusion that IL-25 and IL-33 may act as compensatory alarmins in the absence of TSLPR signaling.

Comment 2:

“Line 200: The authors need to clearly describe that expression of TSLPR, TSLP, and Fc epsilonRI had been reported in human ASM cells but still unknown in murine ASM cells. This is important point not to mislead the readers. The reviewer feels that the end of this paragraph would be the best place where they describe it.”

Response 2:

Revised as suggested. We now state that expression of TSLPR, TSLP, and FcεRI has been reported in human ASM cells, but remains unknown in murine ASM cells.

Comment 3:

“Lines 216/217: The sentence will be read as that “their (IL-33 and IL-25) upregulation in the TSLPR-deficient mice” had been reported already. Is it true?”

Response 3:

The sentence has been rephrased to clarify.

Comment 4:

“Lines 73-75: The authors need to describe that they demonstrated it for human ASM cells (but unknown for mice). “ASM” to be “human ASM” in the line 74. Line 192: “ASM” to be “human ASM cells”. Line 193/194: “likely reflects” to be “might reflect”. Line 197: “ASM cells”to be “human ASM cells”. These specifications are necessary not to mislead the readers.”

Response 4:

All relevant mentions have been changed to “human ASM cells,” and “likely reflects” has been revised to “might reflect.”

Comment 5:

”Line 85: “eight weeks” is wrong. To be changed to “seven weeks”. Line 92: “8 weeks” is wrong. To be changed to “7 weeks”

Response 5:

Corrected.

Comment 6:

“ Fig. 1A: According to the illustration, the last i.n. administration was done on the day 50. Is it correct? In lines 257-260 of the Methods section, the authors described that the administration was for the 7 weeks, and it could be read as that the last administration was done on the day 47. What was done on the day 50? The same amount of i.n. administration with the day 47?”

Response 6:

Clarified. The final HDM administration was performed on day 47, and mice were sacrificed on day 50.

Comment 7:

“Fig. 1B-D: Specify the day number for the data collection. According to the lines 257-260 of the Methods section, it would be the day 52 if the day 50 was the last administration or the day 49 if the day 47 was the last administration.”

Response 7:

Legends revised to specify that lung function data were collected on day 50.

Comment 8:

“Line 114: Is the description correct for the violin plots of the Fig. 2 C, E-H and the scattered plot with not error bars of the Fig. 2D?”

Response 8:

Corrected. The legend now accurately describes the violin plots (Fig. 2C, E–H) and scatter plot (Fig. 2D).

Comment 9:

“Lines 96/97, 114, 132, 146, 168: Which do the last sentences of the Figure legends mean that the authors showed all the data of the 3 independent experiments or data from only 1 selected experiment among the 3 independent experiments with similar results?”

Response 9:

All the data of the 3 independent experiments were included in the figures.

Comment 10: “Line 349: There is no indication with *, ** and *** despite the statistical differences in all the Figures”.

Response 10:

Revised.

---

## [Decision Letter · Decision Letter 2]

8 Oct 2025

Dear Dr. Gounni,

Please submit your revised manuscript by Nov 22 2025 11:59PM. If you will need more time than this to complete your revisions, please reply to this message or contact the journal office at plosone@plos.org . A rebuttal letter that responds to each point raised by the academic editor and reviewer(s). You should upload this letter as a separate file labeled 'Response to Reviewers'.A marked-up copy of your manuscript that highlights changes made to the original version. You should upload this as a separate file labeled 'Revised Manuscript with Track Changes'.An unmarked version of your revised paper without tracked changes. You should upload this as a separate file labeled 'Manuscript'.

We look forward to receiving your revised manuscript.

Kind regards,

Hiroyasu Nakano, M.D., Ph.D.

Academic Editor

PLOS ONE

Journal Requirements:

**Additional Editor Comments:**

Although the revised manuscript is nearly acceptable, the reviewer still has a few minor concerns that should be addressed before it can be accepted.

Reviewers' comments:

Reviewer's Responses to Questions

**Comments to the Author**

Reviewer #1: All comments have been addressed

Reviewer #2: (No Response)

2. Is the manuscript technically sound, and do the data support the conclusions?

Reviewer #1: Yes

Reviewer #2: Yes

3. Has the statistical analysis been performed appropriately and rigorously?

Reviewer #1: Yes

Reviewer #2: No

4. Have the authors made all data underlying the findings in their manuscript fully available?

Reviewer #1: (No Response)

Reviewer #2: Yes

5. Is the manuscript presented in an intelligible fashion and written in standard English?

Reviewer #1: Yes

Reviewer #2: Yes

Reviewer #1: The authors have responded to the comments by the reviewers. I have no additional comments on this paper. I think that it could be accepted for PLoS ONE.

Reviewer #2: The authors responded to the major comment of the Reviewer by measuring TSLP, IL-33 and IL-25 expression in BALF and lung, which strengthened their work. The Reviewer has only minor comments below:

1. Line 358 and all the legends of Figures and Supplementary Figures: Is the description “The data were indicated as mean+/-SEM” correct for the violin plots? Isn’t it to be “median and quartiles” in violin plots by the GraphPad Prism software?

2. Figure 2D: The bars for the means and the SEMs are hard to see.

3. Figure 2: The location of the panels to be improve for readability. The BALF TSLP and IL-33 panels (J and K) to just follow panels for BALF Th cytokines (A-D). Lung TSLP, IL-33 and IL-35 panels (L-N) to just follow panels for lung Th cytokines (E-H).

4. Line 220: “TSLP-IL-33 signaling” to be “TSLP signaling and attenuation of IL-33-signaling”

**Do you want your identity to be public for this peer review?** For information about this choice, including consent withdrawal, please see our Privacy Policy

Reviewer #1: No

Reviewer #2: No

---

## [Author Response · Author response to Decision Letter 3]

11 Oct 2025

Response to Reviewer 2:

We thank the reviewer for taking the time to review our manuscript and for their constructive comments. We have addressed each of the suggestions in detail below and have revised the manuscript accordingly.

Comment 1:

Line 358 and all the legends of Figures and Supplementary Figures: Is the description “The data were indicated as mean ± SEM” correct for the violin plots? Isn’t it to be “median and quartiles” in violin plots by the GraphPad Prism software?

Response:

We thank the reviewer for catching this important detail. You are correct that in GraphPad Prism, violin plots display the median and interquartile range rather than mean ± SEM. We have corrected the figure legends accordingly to accurately reflect the data representation. The revised text now reads:

Comment 2:

Figure 2D: The bars for the means and the SEMs are hard to see.

Response:

We appreciate this observation. We have enhanced the visibility of the mean ± SEM bars in Figure 2D by adjusting line thickness and color contrast in GraphPad Prism to ensure they are clearly visible in both print and digital formats.

Comment 3:

Figure 2: The location of the panels to be improved for readability. The BALF TSLP and IL-33 panels (J and K) to just follow panels for BALF Th cytokines (A-D). Lung TSLP, IL-33, and IL-35 panels (L-N) to just follow panels for lung Th cytokines (E-H).

Response:

Thank you for this helpful suggestion. We have rearranged the panels in Figure 2 to follow the reviewer’s proposed order, improving logical flow and readability.

Reviewer Comment 4:

Line 220: “TSLP-IL-33 signaling” to be “TSLP signaling and attenuation of IL-33 signaling.”

Response:

The phrase has been revised to “TSLP signaling and attenuation of IL-33 signaling” in line 220 to better reflect the mechanistic distinction between these pathways.

---

## [Decision Letter · Decision Letter 3]

15 Oct 2025

TSLPR deficiency attenuates AHR independently of eosinophilia and mucus secretion in chronic HDM mouse model of allergic asthma

PONE-D-25-32039R3

Dear Dr. Gounni,

We’re pleased to inform you that your manuscript has been judged scientifically suitable for publication and will be formally accepted for publication once it meets all outstanding technical requirements.

Kind regards,

Hiroyasu Nakano, M.D., Ph.D.

Academic Editor

PLOS ONE

Additional Editor Comments (optional):

Reviewers' comments:

Reviewer's Responses to Questions

**Comments to the Author**

Reviewer #2: All comments have been addressed

2. Is the manuscript technically sound, and do the data support the conclusions?

Reviewer #2: Yes

3. Has the statistical analysis been performed appropriately and rigorously?

Reviewer #2: Yes

4. Have the authors made all data underlying the findings in their manuscript fully available?

Reviewer #2: Yes

5. Is the manuscript presented in an intelligible fashion and written in standard English?

Reviewer #2: Yes

Reviewer #2: (No Response)

**Do you want your identity to be public for this peer review?** For information about this choice, including consent withdrawal, please see our Privacy Policy

Reviewer #2: No

---

## [Editor Report · Acceptance letter]

PONE-D-25-32039R3

PLOS ONE

Dear Dr. Gounni,

I'm pleased to inform you that your manuscript has been deemed suitable for publication in PLOS ONE. Congratulations! Your manuscript is now being handed over to our production team.

Kind regards,

on behalf of

Professor Hiroyasu Nakano

Academic Editor

PLOS ONE